# Surprisingly long lifetime of methacrolein oxide, an isoprene derived Criegee intermediate, under humid conditions

Yen-Hsiu Lin[1], Cangtao Yin [1], Kaito Takahashi [1] & Jim Jr-Min Lin [1,2✉]

Ozonolysis of isoprene, the most abundant alkene, produces three distinct Criegee intermediates (CIs): $CH_2OO$, methyl vinyl ketone oxide (MVKO) and methacrolein oxide (MACRO). The oxidation of $SO_2$ by CIs is a potential source of $H_2SO_4$, an important precursor of aerosols. Here we investigated the UV-visible spectroscopy and reaction kinetics of thermalized MACRO. An extremely fast reaction of *anti*-MACRO with $SO_2$ has been found, $k_{SO2} = (1.5 \pm 0.4) \times 10^{-10}$ cm$^3$ s$^{-1}$ ($\pm 1\sigma$, $\sigma$ is the standard deviation of the data) at 298 K (150 − 500 Torr), which is ca. 4 times the value for *syn*-MVKO. However, the reaction of *anti*-MACRO with water vapor has been observed to be quite slow with an effective rate coefficient of $(9 \pm 5) \times 10^{-17}$ cm$^3$ s$^{-1}$ ($\pm 1\sigma$) at 298 K (300 to 500 Torr), which is smaller than current literature values by 1 or 2 orders of magnitude. Our results indicate that *anti*-MACRO has an atmospheric lifetime (best estimate ca. 18 ms at 298 K and RH = 70%) much longer than previously thought (ca. 0.3 or 3 ms), resulting in a much higher steady-state concentration. Owing to larger reaction rate coefficient, the impact of *anti*-MACRO on the oxidation of atmospheric $SO_2$ would be substantial, even more than that of *syn*-MVKO.

---

[1] Institute of Atomic and Molecular Sciences, Academia Sinica, Taipei, Taiwan. [2] Department of Chemistry, National Taiwan University, Taipei, Taiwan.
✉email: jimlin@gate.sinica.edu.tw

soprene is the most abundant unsaturated hydrocarbon in the atmosphere[1]. Ozonolysis of isoprene produces three kinds of carbonyl oxides (also called Criegee Intermediates, CIs): $CH_2OO$, methyl vinyl ketone oxide (MVKO: $CH_3C(C_2H_3)OO$), and methacrolein oxide (MACRO: $CH_2=C(CH_3)CHOO$)[2–4]. CIs are very reactive species. In 2012, their reactions with $SO_2$ were found to be faster than previously thought by orders of magnitude[5]. The oxidation of $SO_2$ ($SO_2 \rightarrow SO_3$) has gained wide attention because it is an important process in the formation of secondary aerosols ($SO_3 \rightarrow H_2SO_4$)[6–15]. Field and chamber studies, pioneered by Mauldin et al.[9], indicate that there is a non-OH oxidant contributing to the oxidation of atmospheric $SO_2$ and this new oxidant may be Criegee intermediates[9–15]. However, it is impractical to measure the very reactive CIs in the atmosphere. Their atmospheric concentrations can only be estimated through kinetics analysis. For example, Novelli et al. have given an average estimate of the CI concentration of ca. $5 \times 10^4$ molecules $cm^{-3}$ (with an order of magnitude uncertainty) for the two environments they studied[16]. (For simplicity, we will use '$cm^{-3}$' for the unit of molecular number density, instead of the more formal 'molecules $cm^{-3}$'.) Note that older estimations may have larger uncertainties in the CI concentrations since the related reaction kinetics were not well determined at that time.

On the other hand, laboratory studies on individual CI reactions have revealed that the reactivity (thus the atmospheric fate) of a CI would strongly depend on its structure[17,18]. For $CH_2OO$ and anti-$CH_3CHOO$, which have a H-atom at the syn position, the main decay pathway is their reactions with water vapor ($H_2O$ monomer and dimer)[18–23]. These reactions are extremely fast, resulting in very low steady-state concentrations of such CIs, which are too low to oxidize atmospheric $SO_2$ at any substantial level[17]. For syn-$CH_3CHOO$ and $(CH_3)_2COO$, which have an alkyl group at the syn position, their unimolecular reactions via intramolecular 1,4-H-atom transfer are the major decay process, which also generates OH radicals[24–31]. These unimolecular processes are not slow and essentially limit the steady-state concentrations of such CIs[17,32].

Different from alkyl-substituted CIs, MVKO and MACRO have a C=C double bond, which forms extended conjugation with the carbonyl oxide functional group (resonance stabilized). The pioneering works of Lester and coworkers have demonstrated a photolytic synthesis method that allows direct detection of MVKO and MACRO[33,34]. Via this new synthesis scheme, recent studies have shown that the resonance-stabilization would affect the reactivity and thus the atmospheric fate of MVKO[35,36].

For MVKO, there are four possible isomers (or conformers)[2,33,35,37]. Similar to simpler CIs, the barrier of rotating the carbonyl C=O bond is high, resulting in non-interconverting syn and anti isomers (following the nomenclature of Barbar et al.)[33,38]. However, the barrier of rotating the C—C single bond between the C=C and C=O bonds is low, giving essentially an equilibrium mixture of cis and trans conformers[32,39,40]. It has been predicted that anti-trans-MVKO would quickly interconvert to anti-cis-MVKO ($>10^6$ $s^{-1}$)[32], which decays quickly via fast 1,5-ring closure to form dioxole with a rate coefficient of ca. $2100$ $s^{-1}$ at $298$ K[32,33,41,42]. As a result, anti-MVKO was not observed experimentally, presumably due to short lifetime and/or low yield[35].

Caravan et al. have found that syn-MVKO reacts with $SO_2$ and formic acid as fast as other alkyl CIs do. Furthermore, based on their global chemistry and transport model, they have shown that syn-MVKO could significantly increase the atmospheric oxidation of $SO_2$ and the removal of formic acid, where the isoprene emission is high. The high impact of syn-MVKO is mostly due to the abundance of isoprene (its source) and its slow decay (slow unimolecular decay and slow reaction with water vapor)[35]. The

slow decay of syn-MVKO is related to its resonance-stabilized electronic structure[32,36], which would be disrupted at the transition state of the unimolecular reaction[33,43]. Another interesting aspect of this resonance stabilization is that the iodine-atom adduct of MVKO is relatively less stable compared to the cases of alkyl CIs[36].

MACRO also has a resonance-stabilized electronic structure and two non-interconverting families of conformers (Fig. 1). It has also been predicted that syn-trans-MACRO (following the nomenclature of Vansco et al.)[34] would interconvert quickly ($>10^6$ $s^{-1}$) to syn-cis-MACRO, which would undergo fast unimolecular decay ($k_{uni} = 2500$ $s^{-1}$) to form dioxole, while anti conformers are expected to have slow unimolecular decay (ca. $10$ $s^{-1}$)[32]. These theoretically predicted values are from Vereecken et al.[32] who utilized the structure–activity relationships which considered the best available theoretical and experimental results at that time. However, similar to other anti types of CIs ($CH_2OO$ and anti-$CH_3CHOO$)[17,18], the reaction of anti-MACRO with water vapor was predicted to be fast ($k_{water-eff} = 7.2 \times 10^{-15}$ $cm^3$ $s^{-1}$ by Anglada et al.[44] or $6.3 \times 10^{-16}$ $cm^3$ $s^{-1}$ by Vereecken et al.[32] at relative humidity (RH) = 70% and 298 K, considering both water monomer and dimer reactions). If so, the fast reaction with water vapor (ca. $10^3$ $s^{-1}$) would result in a very low steady-state concentration of anti-MACRO, diminishing its atmospheric impact. Nonetheless, as will be shown later, this picture is incorrect.

Very recently, Vansco et al. have reported the electronic spectroscopy and photochemistry of MACRO; as the authors have mentioned, "This UV–visible detection scheme will enable study of its unimolecular and bimolecular reactions under thermal conditions of relevance to the atmosphere."[34] Following their method, here we prepared MACRO starting from the photolysis of E-1,3-diiodo-2-methylprop-1-ene precursor (Fig. 1). The time-resolved UV–visible spectrum of MACRO was recorded by using a continuous broadband light source and a grating spectrometer equipped with an ultrafast CMOS camera. Analyzing the time series of the spectra allowed us to retrieve the spectrum of MACRO and its time-dependent concentration.

To our surprise, the reaction of MACRO with water vapor was found to be much slower than previous predictions[32,44] by one or two orders of magnitude, implying much longer atmospheric lifetime (ca. 18 ms vs. 3 or 0.3 ms[32,44]) and higher steady-state concentrations for atmospheric MACRO. On the other hand, the resonance-stabilized MACRO still reacts extremely fast with $SO_2$. Based on the results of a recent global chemistry and transportation model of MVKO[35], our data suggest that the impact of

**Fig. 1 Synthesis and conformers of methacrolein oxide (MACRO).** We follow the method and nomenclature of Vansco et al.[34].

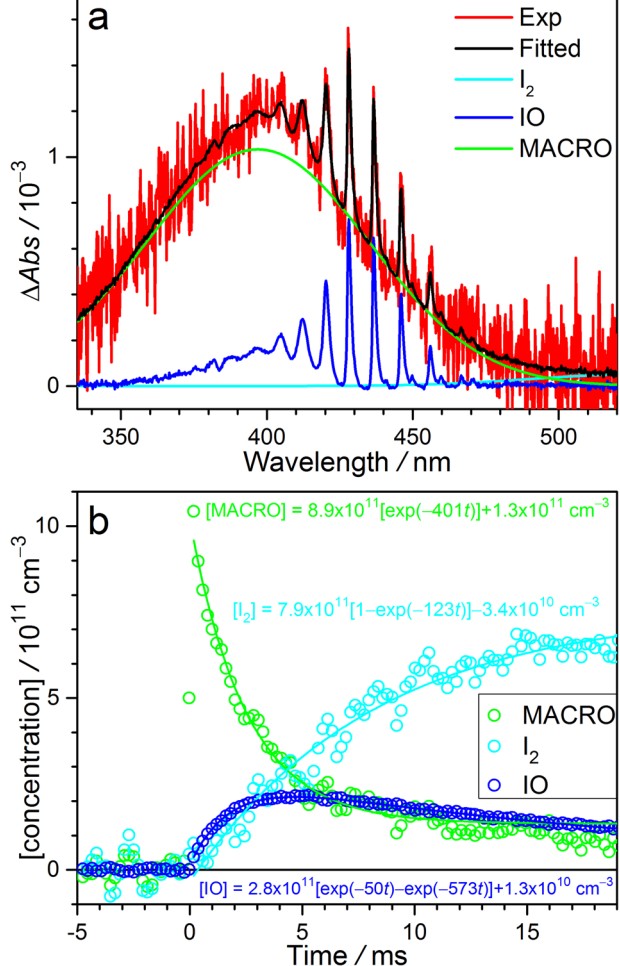

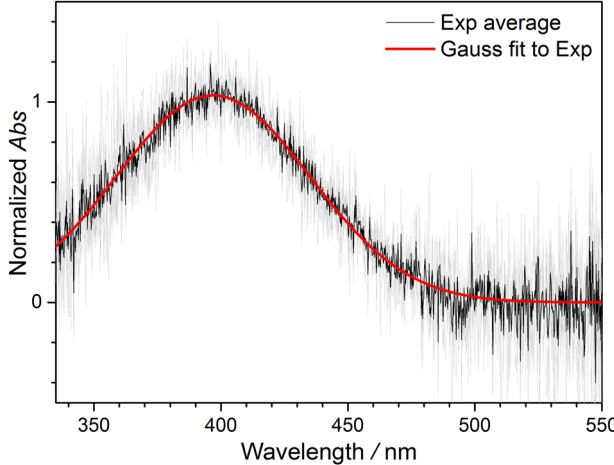

**Fig. 3 Height-normalized spectra of MACRO.** The spectra of MACRO are obtained from the difference spectra between the spectra without $SO_2$ and the spectra with $[SO_2] \cong 10^{14}$ cm$^{-3}$ at 0.18 ms delay time. Four spectra obtained from Exp #1−4 are plotted as gray lines, and their average is plotted as a black line. The red curve shows the Gaussian fit to the average spectrum with peak position 397 nm and full width at half-maximum (FWHM) 77 nm.

**Fig. 2 Absorption spectra and their time evolutions. a** The difference absorption spectrum recorded in the 1,3-diiodo-2-methylprop-1-ene/$O_2$ photolysis system at 298 K and 500 Torr at 0.4 ms photolysis-probe delay time (red line, the data is from Exp #13, see Supplementary Table 2). The absorption contributions of MACRO (green), IO (blue), and $I_2$ (cyan; black is the sum of MACRO, IO, and $I_2$) are also shown. **b** Concentrations of MACRO, IO, and $I_2$ obtained by fitting the spectra of Exp #13 at [$H_2O$] = 0 at each photolysis-probe delay time. [MACRO] is derived by using the reported peak cross section = $3 \times 10^{-18}$ cm$^2$ by Vansco et al.[34], [IO] and [$I_2$] are estimated with literature cross sections[45,46] (see Supplementary Fig. 4).

MACRO on the oxidation of atmospheric $SO_2$ would also be substantial.

## Results and discussion

**Analysis of the observed UV spectrum.** Figure 2a shows the time-resolved difference absorption spectra recorded in the photolysis reactor. Here 'difference' means the change after the photolysis laser pulse. We can see three spectral features in the spectrum: (i) a very broad and structureless absorption band peaking at ca. 397 nm; (ii) absorption of IO which has distinctive sharp peaks between 400 and 460 nm[45]; (iii) broad absorption band of $I_2$ extending to 520 nm[46]. The presence of IO and $I_2$ is similar to previous investigations of $CH_2OO$[47–49], $CH_3CHOO$[23,50], $(CH_3)_2COO$[51], and MVKO[35,36].

**UV absorption spectrum of MACRO.** Under the same experimental conditions, $SO_2$ was added to scavenge CIs (see Supplementary Fig. 1). We found that the spectral feature (i) disappears, indicating its spectral carrier is a Criegee intermediate. We

further subtracted the time-resolved spectra recorded at [$SO_2$] = $1 \times 10^{14}$ cm$^{-3}$ from those without adding $SO_2$ at each photolysis-probe delay time. This operation removed most of the absorption signals of IO and $I_2$ (and also other minor byproducts), and the resulted spectra would be mainly from the Criegee intermediate. Considering that we were using the same precursor and preparation method of Vansco et al.[34], we assigned this CI to MACRO.

The spectrum of MACRO can be well fitted with a Gaussian function (Fig. 3). This spectrum is similar to that of Vansco et al. who reported a broad spectrum of MACRO peaked at 380 nm with weak oscillatory structure at long wavelengths ascribed to vibrational resonances[34]. However, we do not observe such oscillatory structure; the differences are presumably due to different temperatures of the MACRO samples (thus, the conformer populations may be different), as Vansco et al. recorded their spectrum under a jet cool condition[34].

We may decompose the observed spectra into the contributions of MACRO, IO, and $I_2$ (Fig. 2a and Supplementary Fig. 4) with a least-squares fit. The resulted signal intensities (converted to concentrations) of MACRO, IO, and $I_2$ are plotted in Fig. 2b as a function of the delay time. The intensities of IO and $I_2$ grow with time, indicating they are secondary products. While the kinetics of IO and $I_2$ formation may be interesting, we like to focus on MACRO in this work. We can see that the lifetime of MACRO in this particular experiment is ca. 3 ms, much longer than the predicted value for *syn*-MACRO (<0.4 ms, based on its $k_{uni} = 2500$ s$^{-1}$[32]; and other chemical processes would further shorten the lifetime). Therefore, we conclude that the observed spectral carrier should be *anti*-MACRO, similar to the case of MVKO[35]. Note that the long lifetime conformer of MVKO is *syn*-MVKO (following the nomenclature of Barbar et al.)[33] which has a structure similar to *anti*-MACRO. For simplicity, we will use MACRO to represent *anti*-MACRO in the following analysis.

**Kinetics of MACRO reaction with $SO_2$.** The above analysis has been repeated for experiments adding various [$SO_2$]. The resulted MACRO signal intensities at each photolysis-probe delay time are plotted in Fig. 4a. The decay of MACRO signal can be fitted with a single exponential function to yield a pseudo-first-order rate

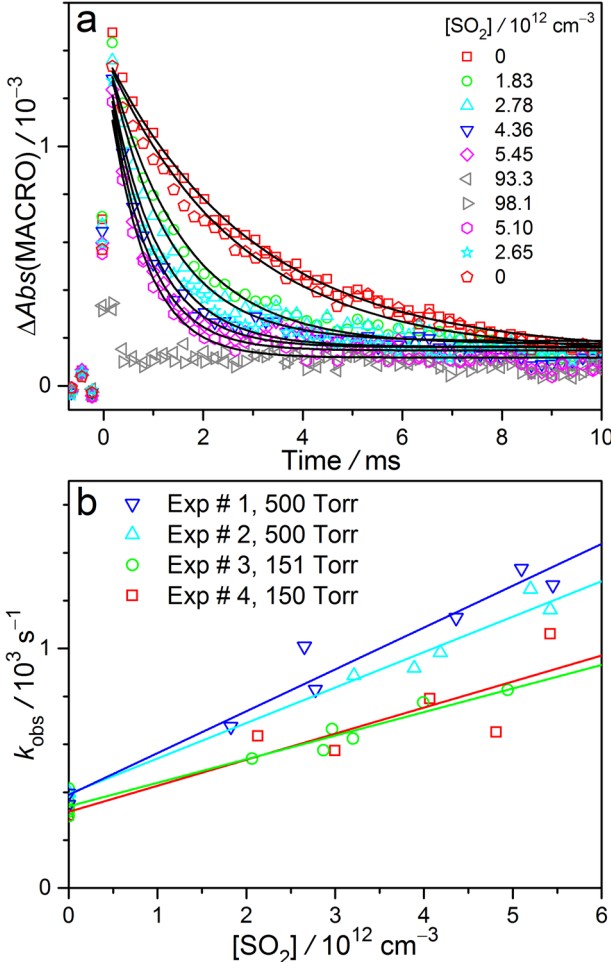

**Fig. 4 Kinetic plots of MACRO + SO₂. a** Time profiles of the absorbance changes of MACRO obtained by fitting the spectra at each delay time under various [SO₂] at 298 K and 500 Torr. The data are from Exp #1. The black curves show the fitting results of single exponential decay. **b** Pseudo-first-order plot of the reaction rate coefficients of MACRO with SO₂ at 298 K.

coefficient, $k_{obs}$, at each [SO₂].

$$\Delta Abs(\text{MACRO}) = \sigma L[\text{MACRO}](t) = \sigma L[\text{MACRO}]_0 \exp(-k_{obs}t)$$

where $\sigma$ is the absorption cross section of MACRO[34] and $L$ is the optical path length (when reporting $\Delta Abs(\text{MACRO})$, we use its peak value at 397 nm). Figure 4b shows that $k_{obs}$ increases linearly with [SO₂].

$$k_{obs} = k_0 + k_{SO2}[\text{SO}_2]$$

The slope would correspond to the rate coefficient $k_{SO2}$ of the bimolecular reaction of MACRO with SO₂, while the intercept $k_0$ would account for other decay processes of MACRO that are independent on [SO₂], like reactions with radical byproducts (including MACRO self-reaction), unimolecular decay, etc.

Because that MACRO can be fully scavenged within 0.3 ms under a high [SO₂] ($\geq 9.3 \times 10^{13}$ cm⁻³), we may further improve the analysis by subtracting the high [SO₂] spectrum (the average spectrum at the two highest [SO₂]) from other low [SO₂] spectra to remove most of the byproduct contributions while some minor amounts of IO and I₂ may still remain (SO₂ scavenge method). The resulted spectra were then decomposed into the contributions of MACRO and IO and I₂. The time profiles of MACRO are plotted in Supplementary Fig. 2. When using this SO₂ scavenge method, we

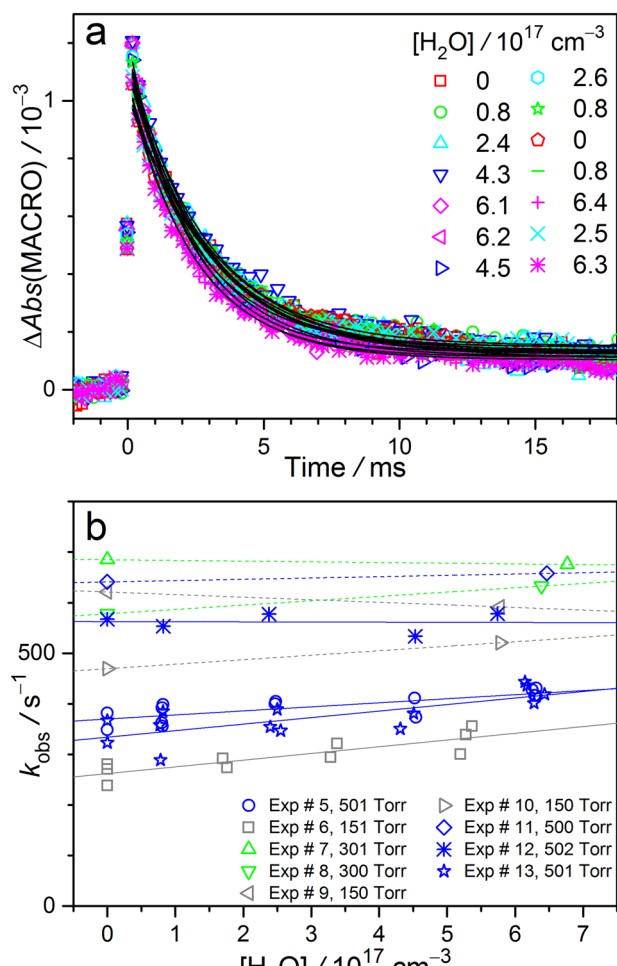

**Fig. 5 Kinetic plots of MACRO + H₂O. a** Time profiles of the MACRO absorption signal obtained by fitting the spectra at each delay time under various [H₂O] at 298 K and 501 Torr. The data are from Exp #13 (Supplementary Table 2). The black curves show the fitting results of a single exponential decay function. **b** The pseudo-first-order plot of the reaction rate coefficients of MACRO with H₂O at 298 K.

did not include the data point at the first delay time (0.18 ms) due to incomplete scavenging.

The kinetic results of MACRO + SO₂ reaction are summarized in Supplementary Table 1. The data at 150 and 500 Torr do not show significant difference after considering the experimental uncertainties. For these four sets of experimental data, we report the rate coefficient to be $(1.5 \pm 0.4) \times 10^{-10}$ cm³ s⁻¹ at 298 K and 150–500 Torr ($\pm 1\sigma$, $\sigma$ is the standard deviation of the data). The rate coefficients of SO₂ reactions with other CIs (CH₂OO[5,52], *anti*- and *syn*-CH₃CHOO[21,23], (CH₃)₂COO[30,51], MVKO[35,36]) are in the range of $(0.4–2.2) \times 10^{-10}$ cm³ s⁻¹[17,18]. It appears that the resonance-stabilization of *anti*-MACRO does not reduce its reactivity towards SO₂.

**Kinetics of MACRO reaction with H₂O.** The same method has been applied to investigate the kinetics of MACRO reaction with water vapor. To our surprise, the effect of water in the decay of MACRO is rather weak as shown in Fig. 5, indicating slow reaction. From the plots of $k_{obs}$ as a function of [H₂O] (Fig. 5b), we can see that the slopes are quite insignificant; some of them are even negative, indicating that these rate measurements are close to our measurement limit (see Supplementary Table 2). Note that the highest [H₂O] used is ca. $6 \times 10^{17}$ cm⁻³ (ca. 18

**Table 1 Summary of theoretically estimated unimolecular rate coefficient $k_{uni}$ and bimolecular rate coefficients of the reactions with water monomer and dimer ($k_{H2O}$ and $k_{(H2O)2}$) at 298 K for selected CIs.**

| Relative humidity | | | | 70% | | 35% | | |
|---|---|---|---|---|---|---|---|---|
| [H₂O]/10¹⁷ cm⁻³ | | | | 5.4 | | 2.7 | | |
| [(H₂O)₂]/10¹⁴ cm⁻³ | | | | 6.7 | | 1.7 | | |
| CI | $k_{uni}$ (s⁻¹) | $k_{H2O}$ (cm³ s⁻¹) | $k_{(H2O)2}$ (cm³ s⁻¹) | $k_{water\text{-}eff}$ ᵃ (cm³ s⁻¹) | $k_{atm}$ ᵇ (s⁻¹) | $k_{water\text{-}eff}$ ᵃ (cm³ s⁻¹) | $k_{atm}$ ᵇ (s⁻¹) | Ref. |
| syn-MVKO | 33 | $3.4 \times 10^{-19}$ | $9.2 \times 10^{-16}$ | $1.5 \times 10^{-18}$ | 34 | $9.1 \times 10^{-19}$ | 33 | 35,44 |
| | 50 | $8.1 \times 10^{-20}$ | $3.1 \times 10^{-16}$ | $4.6 \times 10^{-19}$ | 50 | $2.7 \times 10^{-19}$ | 50 | 32 |
| | [70 ± 15] | – | – | — | 70 | — | 70 | 43 |
| anti-MACRO | – | $3.0 \times 10^{-15}$ | $3.4 \times 10^{-12}$ | $7.2 \times 10^{-15}$ | 3900 | $5.1 \times 10^{-15}$ | 1400 | 44 |
| | 10 | $1.9 \times 10^{-16}$ | $3.6 \times 10^{-13}$ | $6.3 \times 10^{-16}$ | 350 | $4.1 \times 10^{-16}$ | 120 | 32 |
| | < 25 | $4.9 \times 10^{-17}$ | $1.6 \times 10^{-14}$ | $6.9 \times 10^{-17}$ | < 62 | $5.9 \times 10^{-17}$ | < 41 | Theoryᶜ |
| | 7ᵈ | | | [$(9 \pm 5) \times 10^{-17}$] | 56 | [$(9 \pm 5) \times 10^{-17}$] | 31 | This work |
| syn-MACRO | — | $2.3 \times 10^{-19}$ | $2.4 \times 10^{-15}$ | $1.3 \times 10^{-17}$ | – | $1.7 \times 10^{-18}$ | — | 44 |
| | 2500 | $1.5 \times 10^{-19}$ | $5.6 \times 10^{-16}$ | $8.4 \times 10^{-19}$ | 2500 | $5.0 \times 10^{-19}$ | 2500 | 32 |
| | 2600 | $1.4 \times 10^{-20}$ | $4.1 \times 10^{-17}$ | $6.4 \times 10^{-20}$ | 2600 | $3.9 \times 10^{-20}$ | 2600 | Theoryᶜ |

Available experimental results are shown in square brackets.
ᵃ$k_{water\text{-}eff} = (k_{H2O}[H_2O] + k_{(H2O)2}[(H_2O)_2])/[H_2O]$ at 298 K, in which [(H₂O)₂] is estimated with $K_{eq} = P_{dimer}/P_{monomer}^2 = 0.0556$ bar⁻¹ reported by Anglada et al.[53]
ᵇ$k_{atm} = k_{uni} + k_{water\text{-}eff}[H_2O]$.
ᶜThis work, based on QCISD(T)/CBS//B3LYP/6-311+G(2d,2p) (CBS = complete basis set extrapolated).
ᵈOur best estimated theoretical value (see text).

Torr), which has replaced a larger portion (18/150 = 12%) of the bath gas if the total pressure is only 150 Torr (N₂ balance). Thus, we think it may require some cautions to view the data of 150 Torr, because the reaction environment (type of bath gas) changes at various [H₂O]. Nonetheless, no trend can be found for pressures from 150 to 500 Torr. Finally, we chose the weighted average from six experimental sets (300 and 500 Torr and 298 K, Supplementary Table 2), to report the effective rate coefficient for the reaction of MACRO with water vapor, $k_{water\text{-}eff} = (9 \pm 5) \times 10^{-17}$ cm³ s⁻¹ (±1σ). As mentioned above, we are not confident enough to determine the lower limit of $k_{water\text{-}eff}$.

**MACRO isomers.** As pointed out by Vereecken et al., the cis–trans interconversion (Supplementary Fig. 6) (>10⁶ s⁻¹) is orders-of-magnitude faster than other chemical processes, such that the cis and trans conformers will be in near-equilibrium and should be considered as a single pool of species[32]. Following this idea, we summarize the rate coefficients of the unimolecular processes and reactions with water vapor (monomer and dimer[53]) in Table 1 for relevant CIs[32,35,43,44].

As shown in Table 1, the predicted unimolecular decay rates are very different for syn- and anti-MACRO. Syn-MACRO would have a rather short lifetime of ca. $1/2500 = 4 \times 10^{-4}$ s[32], which means its steady-state concentration would be very low. The experimental lifetime of MACRO is found to depend on the signal intensity—the higher the signal is, the shorter the lifetime. This is due to the fact that the inevitable reactions of MACRO with radical byproducts, including I atoms, IO radicals, MACRO itself and the products from the fast decomposition of syn-MACRO (similar to the case of anti-MVKO)[42] would shorten its lifetime. Supplementary Fig. 3 shows the plot of $k_{obs}$ against [MACRO]₀. The linear relationship supports the above mechanism. Extrapolating $k_{obs}$ to zero [MACRO]₀ would effectively remove the bimolecular contributions and give an estimate for the unimolecular lifetime of MACRO. The preliminary data of Supplementary Fig. 3 are consistent with $k_{uni} \cong k_{obs}([MACRO]_0 = 0) < 50$ s⁻¹, which gives a lifetime > 20 ms. This long-lived MACRO cannot be syn-MACRO. Thus, the observed signal should belong to anti-MACRO.

**Compare of $k_{water\text{-}eff}$ with previous theory.** The value of $k_{water\text{-}eff}$ of anti-MACRO (Table 1) is smaller than those of CH₂OO and

anti-CH₃CHOO[17,18] by orders of magnitude, suggesting that the extended conjugation of anti-MACRO correlates with the lower reactivity towards water vapor (monomer and dimer), since the alkyl-substituted CIs lack the resonance-stabilized electronic structure of the extended conjugation.

Anglada et al. have predicted the rate coefficients for anti-MACRO reactions with water monomer and dimer, giving $k_{water\text{-}eff} = 7.2 \times 10^{-15}$ cm³ s⁻¹ at RH = 70% and 298 K (Table 1)[44]. To our surprise, this value is ca. 80 times larger than our experimental value. However, Vereecken et al. have pointed out that the level of theory used by Anglada et al.[44] tends to underestimate the barriers for the CI reactions with water monomer and dimer[32]. Using 'structure–activity relationship', Vereecken et al. scaled the barrier heights of a number of CI reactions by considering the best known theoretical and experimental data (mainly for CH₂OO and anti-/syn-CH₃CHOO) at that time[32]. The resulted rate coefficients of MACRO are also shown in Table 1. We can see this 'scaling' does reduce the gap (from 80 times to 7 times) between the theoretical predictions and our experimental data. Since the reference data in the work of Vereecken et al.[32] do not contain trustable data for reactions of CIs having a conjugated C=C group (i.e., there is no good anchor point for the scaling), this difference may be reasonable. Also note that Vereecken et al.[32] have estimated an uncertainty of one order of magnitude for their rate coefficients at 298 K.

With details given in Supplementary Note 3, we found that it is important to calculate the reaction barrier heights at a high level of quantum chemistry theory like QCISD(T)/CBS//B3LYP/6-311+G(2d,2p) (CBS = complete basis set extrapolation)[54–60]. For example, the QCISD(T)/CBS barriers are ca. 1.4 or 2.0 kcal mol⁻¹ higher than those calculated at QCISD(T)/aug-cc-pVTZ (AVTZ) for various reactions between (CH₂=CH)CHOO conformers with H₂O monomer or dimer (Supplementary Fig. 7 and Supplementary Table 4), indicating that only using the AVTZ barrier heights would overestimate the reaction rates significantly.

After properly scaling the effect of the basis sets (CBS vs. AVTZ) by using the results of (CH₂=CH)CHOO, which has a structure similar to MACRO, as an anchor point, (Supplementary Fig. 7)[54], our calculation (Table 1) also predicts slower rates compared to previous ones.

**Compare of $k_{\text{water-eff}}$ with ozonolysis experiment.** Newland et al. have analyzed the effect of water vapor in the system of isoprene ozonolysis; in their two-CI model, the isoprene-derived non-$CH_2OO$ CI (sum of MVKO and MACRO) has an effective reaction rate coefficient with water vapor of $(1.1 \pm 0.27) \times 10^{-15}$ cm$^3$ s$^{-1}$[61]. While all conformers of MVKO are expected to react with water vapor much slower ($k_{\text{water-eff}} \leq 10^{-17}$ cm$^3$ s$^{-1}$)[44], the value of Newland et al. is much larger than our $k_{\text{water-eff}}$ for MACRO. At the time (2015) when the work of Newland et al. was published, the knowledge of the reaction kinetics of MVKO and MACRO was not available at all. As multiple CIs are produced in the isoprene ozonolysis system, the kinetics is rather complicated, especially when these CIs have very different reactivities towards water vapor. For example, $CH_2OO$, which has the predominant yield in the isoprene ozonolysis system[2,3], would be quickly consumed by its reaction with water vapor, but MVKO and MACRO would not. See Supplementary Note 2 for an alternative analysis to fit the data of Newland et al.[61]. In fact, Newland et al. have mentioned that the competing effects of the different kinetics of two distinct forms (*syn* and *anti* conformers) in the system may effectively lead to one masking the other under the experimental conditions applied[61].

**Best estimation of $k_{\text{uni}}$.** It is very difficult to experimentally measure the very slow rate of the *anti*-MACRO unimolecular reaction. While our preliminary experimental data (Supplementary Fig. 3) suggest that the unimolecular reaction is slow, we cannot nail down the value of $k_{\text{uni}}$ by the experimental results. On the theoretical side, the unimolecular reaction of *anti*-MACRO proceeds through the OO bending channel forming dioxirane[32,34,39], similar to that of $CH_2OO$[41,62]. By comparing with the results of high-accuracy extrapolation protocols like HEAT-345(Q)[62] or high-level multireference methods like MRCI+Q (Davidson correction)/CBS[41], Yin and Takahashi have found that the QCISD(T)/CBS method slightly underestimates the barrier height of this channel (by ca. 0.4 or 1.2 kcal mol$^{-1}$, respectively) for $CH_2OO$[41]. Our analysis in Supplementary Note 3 shows that for the MACRO unimolecular reaction, the electronic energy obtained by QCISD(T)/CBS would consistently underestimate the barrier height and other factors in the rate calculation, like hindered-rotor partition function calculation and tunneling correction, have very minor effects compared to that of the electronic energy. Therefore, our theoretical value (25 s$^{-1}$) of $k_{\text{uni}}$ of *anti*-MACRO would only represent an upper limit.

Assuming such underestimation in the barrier heights (0.4 or 1.2 kcal mol$^{-1}$) is similar for the unimolecular reactions of MACRO and $CH_2OO$, we may have an overestimation of a factor of 2 or 7 for the reaction rate coefficient at 298 K. Thus, we think the best estimated $k_{\text{uni}}$ at 298 K would be ca. $25/(2 \times 7)^{0.5} = 7$ s$^{-1}$ (the uncertainty may be up to a factor of 3), which is (almost) coincident with the theoretical value of 10 s$^{-1}$ by Vereecken et al. (claimed uncertainty is ca. a factor of 5 for non-H-migration reactions)[32]. Although the uncertainty is still not very small, "for many assessments, however, it is sufficient to determine whether the reaction is significantly faster or slower than competing reactions", mentioned by Vereecken et al.[32].

**Atmospheric lifetime.** Because the unimolecular decay and reaction with water vapor are the predominant processes that determine the atmospheric lifetime of a CI[17,18,32], we may estimate the effective decay rate coefficient $k_{\text{atm}}$ for atmospheric *anti*-MACRO.

$$k_{\text{atm}} = k_{\text{uni}} + k_{\text{water-eff}}[H_2O]$$

Taking the best estimated $k_{\text{uni}}$ (7 s$^{-1}$) and our experimental data of $k_{\text{water-eff}}$ (Table 1), we have $k_{\text{atm}} = 56$ s$^{-1}$ (or <74 s$^{-1}$, if taking our theoretical upper limit of 25 s$^{-1}$ for $k_{\text{uni}}$) for *anti*-

MACRO at RH = 70% and 298 K. Note that the water reaction may still predominate in the decay processes of atmospheric *anti*-MACRO under humid conditions that are typical for tropical forests where the isoprene emission is large. And this atmospheric lifetime (ca. 18 ms, best estimate) is much longer than previously thought (0.3 or 3 ms, see Table 1), indicating that the atmospheric concentration of *anti*-MACRO would be much higher than previously expected. If using the upper limits of $k_{\text{uni}}$ (25 s$^{-1}$) and $k_{\text{water-eff}}[H_2O]$ ($49 + 54 = 103$ s$^{-1}$, $2\sigma$ upper bound, at RH = 70% and 298 K), we then have $k_{\text{atm}} < 128$ s$^{-1}$, which would correspond to a lifetime longer than 7.8 ms.

**Impact of *anti*-MACRO on the oxidation of atmospheric $SO_2$.** This would depend on three factors: (i) the yield of *anti*-MACRO in the ozonolysis of atmospheric alkenes (mainly isoprene), (ii) the atmospheric lifetime of *anti*-MACRO, and (iii) the rate coefficient of *anti*-MACRO reaction with $SO_2$. Each factor is discussed below.

First, based on the recent analysis of Nguyen et al., *anti*-MACRO has a yield of 15% among all stabilized CIs in isoprene ozonolysis, which is very similar to that of *syn*-MVKO (14%)[2,17]. In addition, an earlier study of Zhang and Zhang has shown that the activation energies of $O_3$ cycloaddition to the two double bonds of isoprene are comparable and the barrier heights from the primary ozonides to *syn*-MVKO and *anti*-MACRO are also similar, implying that the *syn*-MVKO and *anti*-MACRO pathways are both accessible[4].

Second, given that *syn*-MACRO and *anti*-MVKO have much shorter lifetimes ($\tau < 1$ ms), the oxidation of atmospheric $SO_2$ by the C4 CIs from isoprene ozonolysis would be mainly by *anti*-MACRO and *syn*-MVKO ($\tau > 10$ ms)[32,35]. The order of magnitude of $k_{\text{atm}}$ of *anti*-MACRO (56 s$^{-1}$) is comparable to that of *syn*-MVKO ($k_{\text{atm}} \cong k_{\text{uni}} = 33$ s$^{-1}$ or 70 s$^{-1}$[33,43]; $k_{\text{water-eff}} \sim 10^{-18}$ cm$^3$ s$^{-1}$)[32,35]. Combined with their similar yields in the isoprene ozonolysis, this suggests that *anti*-MACRO and *syn*-MVKO would have similar steady-state concentrations ([*anti*-MACRO]$_{\text{ss}}$ ≈ [*syn*-MVKO]$_{\text{ss}}$) in the troposphere.

Finally, the rate coefficient of $SO_2$ reaction with *anti*-MACRO, $(1.5 \pm 0.4) \times 10^{-10}$ cm$^3$ s$^{-1}$, is larger than that with *syn*-MVKO, $(4.0-4.2) \times 10^{-11}$ cm$^3$ s$^{-1}$[35,36], by a factor of ca. 4. Overall, the oxidation rate of $SO_2$ by *anti*-MACRO would be larger than that by *syn*-MVKO by a factor of ([*anti*-MACRO]$_{\text{ss}}$/[*syn*-MVKO]$_{\text{ss}}$) ($k_{\text{SO2\_anti-MACRO}}/k_{\text{SO2\_syn-MVKO}}$). This factor would be larger than unity, if we assume that these two CIs are mainly from isoprene ozonolysis with similar yields.

Although $CH_2OO$ has the highest yield (ca. 58%) among the stabilized CIs produced in the ozonolysis of isoprene[2,17], its fast reaction with water vapor results in a rather short atmospheric lifetime (<1 ms)[17-20], too short for $CH_2OO$ to reach any substantial concentration to oxidize atmospheric $SO_2$. Recently Caravan et al., who utilized the up-to-date data of MVKO kinetics, show that *syn*-MVKO has the largest modeled steady-state concentration among all stabilized CIs globally (33% by molecules, 49% by weight)[35]. The above analysis shows that *anti*-MACRO would have similar concentrations as those of *syn*-MVKO and an even larger impact on the $SO_2$ oxidation.

## Conclusion

Following the method of Vansco et al.[34], MACRO has been synthesized and its UV–visible spectroscopy and reaction kinetics have been investigated. Similar to MVKO, MACRO has two non-interconverting isomers, *syn* and *anti* forms. *Syn*-MACRO would undergo fast 1,5-ring closure with a predicted thermal lifetime of <0.4 ms. In our experiments, a much longer lifetime ($\tau > 4$ ms) has been observed, indicating that the spectral carrier is *anti*-

MACRO. The rate coefficient of *anti*-MACRO reaction with $SO_2$ has been determined to be $(1.5 \pm 0.4) \times 10^{-10}$ cm$^3$ s$^{-1}$ at 298 K, which is substantially larger than that of the *syn*-MVKO + $SO_2$ reaction. However, the reaction of *anti*-MACRO with $H_2O$ was found to be quite slow with an effective rate coefficient of $(9 \pm 5) \times 10^{-17}$ cm$^3$ s$^{-1}$ at 298 K, which is smaller than previous theoretical values by 1 or 2 orders of magnitude. Theoretical calculations that properly treat the effect of the conjugated C=C substitution may reproduce the experimental trend.

A recent global chemistry and transport modeling based on the most up-to-date knowledge of MVKO chemistry has shown that *syn*-MVKO is important in the tropospheric processes of $SO_2$ oxidation and formic acid removal[35]. Our results indicate that *anti*-MACRO has an atmospheric lifetime similar to that of *syn*-MVKO, resulting in a similarly substantial steady-state concentration. Combined with the larger rate coefficient of its reaction with $SO_2$, the impact of *anti*-MACRO on the oxidation of atmospheric $SO_2$ would be larger than (at least comparable to) that of *syn*-MVKO.

As mentioned above, to serve as an efficient oxidant of $SO_2$, it is required to have a long-enough lifetime under atmospherically relevant conditions. As shown above and in the literature[35], a resonance-stabilized electronic structure plays an interesting role for CIs. It reduces the reactivity for unimolecular decay and reactions with water vapor, but not for the reactions with $SO_2$. Thus, having a resonance-stabilized electronic structure may be a new direction for searching for a long-lived CI that is able to oxidize atmospheric $SO_2$.

## Methods

**MACRO preparation**. The experimental setup has been published[19,36]. We prepared MACRO following Vansco et al.: ICHC(CH$_3$)CH$_2$I (1,3-diiodo-2-methyl-prop-1-ene, Accela, 97.8% by gas chromatography) + $h\nu$ (248 nm) → CH$_2$=C(CH$_3$)CHI + I, CH$_2$=C(CH$_3$)CHI + O$_2$ → CH$_2$=C(CH$_3$)CHOO + I (Fig. 1)[34]. The precursor concentrations were determined by its UV absorption spectra; the absolute cross sections (Supplementary Fig. 4) have been determined by measuring the weight loss of the precursor sample and the volume flow rate of the dilution gas[63,64] (see Supplementary Note 1).

**CMOS camera spectrometer**. A grating spectrometer (Andor SR303i) and fast CMOS camera (Andor, Marana-4BU11) were used to obtain the time-resolved absorption spectra of the reaction system. A series of spectra (exposure time 0.21 ms (or 0.43 ms) each) were recorded for every photolysis event. The spectrum taken before the photolysis laser pulse was used as the reference spectrum; therefore, the change of absorbance caused by the photolysis laser pulse was recorded transiently. Accumulation of 256, 512 (0.43 ms exposure time), or 1280, 2560 (0.21 ms exposure time) laser pulses was performed to improve the signal-to-noise ratio.

A background spectrum (without adding the MACRO precursor 1,3-diiodo-2-methylprop-1-ene) was recorded under the same experimental condition. This background was due to the interaction between the photolysis laser beam and the used optics (mainly the long-pass filters that reflected the photolysis laser beam and transmitted the probe beam). All the reported spectra are background corrected.

**Theoretical calculations**. We optimized the reactant and transition state geometries on the singlet ground electronic state using B3LYP/6-311+G(2$d$,2$p$)[54,58,59]. See Supplementary Data 1 for the optimized XYZ geometries. The electronic energies were corrected at QCISD(T)/CBS level[54–60], except for the transition states of MACRO + 2H$_2$O, of which the energies were estimated with a correction method detailed in Supplementary Note 3 (Supplementary Figs. 7–9, Supplementary Tables 4–6). The rate coefficients were calculated using the conventional transition state theory method using rigid rotor harmonic oscillator approximations including tunneling correction.

## Data availability

The data supporting the findings of this study are available within the article, its Supplementary Information, and Supplementary Data 1 (XYZ geometries), and from the corresponding author upon reasonable request.

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

## Acknowledgements

This work is supported by Academia Sinica (AS-CDA-106-M05) and Ministry of Science and Technology, Taiwan (MOST 109-2113-M-001-027-MY3 (JJML); MOST 109-2113-M-001-008 (KT); MOST 109-2639-M-009-001-ASP).

## Author contributions

J.J.-M.L. conceived the experiment. Y.-H.L. set up the experiment, performed the measurements, and analyzed the experimental data. C.Y. and K.T. performed the theoretical calculations. Y.-H.L., C.Y., K.T. and J.J.-M.L. discussed the results. J.J.-M.L. wrote the paper.

## Competing interests

The authors declare no competing interests.
