## [Peer Review File · Communications Chemistry]

Reviewers' comments:

Reviewer #1 (Remarks to the Author):

The authors provide experimental data and supporting quantum chemical calculations establishing the rate constant for the reaction of anti methacrolein oxide (MACRO) with SO₂ and H₂O. Given the abundance of isoprene in the atmospheric and the slow unimolecular decay of anti MACRO, the reaction is highly significant. The authors have presented their results clearly and provide evidence of careful data analysis. This sound and fundamentally important study will advance atmospheric chemistry and merits publication. I have only two issues for the authors to consider: (1) They should think more amply about the discrepancy between their MACRO + H₂O rate constant and the isoprene-derived CI + H₂O rate constant reported by Newland et al. (Phys Chem Chem Phys 2015). It is not enough to note that the majority of CI produced from isoprene ozonolysis is CH₂OO. (2) The authors should provide a little justification for the soundness of QCISD(T) theory for estimating reaction barriers and B3LYP for geometries and frequencies.

Reviewer #2 (Remarks to the Author):

Editorial comment: Please see attached file.

Reviewer #3 (Remarks to the Author):

Review of "Surprisingly long lifetime of methacrolein oxide, an isoprene derived Criegee intermediate, under humid conditions"

This study focuses on the determination of the reaction of the anti-MACRO SCI with water, SO₂ and its unimolecular loss rate. It is found that the effective reaction with H₂O is slower than previously reported from theoretical calculations which increases the lifetime of the anti-MACRO SCI from less than a ms to up to 20 ms. The reaction of the anti-MACRO SCI with SO₂ is also found to be very fast and can contribute to the formation of H₂SO₄ due to the longer lifetime of this SCI.

This study is well performed with a solid experimental and theoretical component. The results are of high interest for the community and I fully support the publication after the following points are considered.

Abstract. When talking about the water rate coefficient I would specify it is the effective one as highlighted further in the paper. I would also add the newly calculated lifetime and what was the previously thought value.

Introduction. At the end of the first paragraph, when discussing the importance of SCI as oxidants of SO₂ from previous studies it is important to distinguish between older studies which estimated a much larger SCI concentrations from more recent studies where a more complete body of reactions for SCI was included. At the end of the second paragraph also the study by Novelli et al. (2017) should be included as the first highlighting low expected steady state concentrations in the atmosphere. On page 5 when discussing previous values for the reaction of anti-MACRO I would recommend to highlights the values from the study by Vereecken et al. (2017) as they are correctly accounting for the interconversion of the two conformers and apply the correct barrier height. Indeed, the difference between the measured k effective and the theoretical one from the study by Vereecken et al. (2017) is less than a factor of 10, well within the uncertainties. On page 6, in the last paragraph, similarly as for the abstract, I would specify what lifetime for MACRO was predicted in the atmosphere and from which studies.

Table 1. I would add the experimental result for the unimolecular decomposition of anti-MACRO of $< 50 \text{ s}^{-1}$ as it is consistent with what found with the theory. I would also add the values from the study by Vereecken et al. (2017) for the syn-MVKO. In the caption below the table I would also add the water and water dimers concentrations. Is RH=75% and T=298K a representative set for

isoprene dominated environments such as Amazonia? It could make sense to have two scenarios at higher and lower values and see the bracket of lifetime for the upper and lower limit. Discussion. On page 16 it is underlined that the measured unimolecular must be smaller than 50 s^{-1} implying that the lifetime will be then larger than 20 ms. Although this is technically correct, it is not the full picture as the loss with water and water dimers contribute about 50% to the total loss of MACRO (even more if the unimolecular rate from the study by Vereecken et al. (2017), 10 s^{-1} , is used).

Atmospheric lifetime. Why is the value from the study by Vereecken et al. (2017) used and not the value as calculated within this study? Also, the theoretical value also carries some uncertainty, shouldn't also be considered in the calculation? And which studies reported a lifetime of less than 1 ms?

Impact of anti-MACRO on the oxidation of atmospheric SO_2 . Which other species, apart from isoprene, produce anti-MACRO?

Conclusion. The last line is interesting, but I wonder what is the point of it? Either atmospherically relevant VOCs produce SCI which are long lived or, from an atmospherically point of view, the MACRO might well be the only "long" lived SCI. Which is very interesting on its own.

References

Novelli, A., Hens, K., Tatum Ernest, C., Martinez, M., Nölscher, A. C., Sinha, V., Paasonen, P., Petäjä, T., Sipilä, M., Elste, T., Plass-Dülmer, C., Phillips, G. J., Kubistin, D., Williams, J., Vereecken, L., Lelieveld, J., and Harder, H.: Estimating the atmospheric concentration of Criegee intermediates and their possible interference in a FAGE-LIF instrument, *Atmos. Chem. Phys.*, 17, 7807-7826, doi:10.5194/acp-17-7807-2017, 2017.

Vereecken, L., Novelli, A., and Taraborrelli, D.: Unimolecular decay strongly limits the atmospheric impact of Criegee intermediates, *Phys Chem Chem Phys*, 19, 31599-31612, doi:10.1039/C7CP05541B, 2017.

The manuscript describes an interesting work from a well-known group on atmospheric-relevant kinetics of *anti*-methacrolein oxide (MACRO), which is resonance-stabilized that often leads to an increased radical stability and potentially lower reactivity. To say that, this reviewer feels that authors have over-interpreted or overused their not-so-accurate results too much. This reviewer thinks that probably the main deficiency of this work is rather weak determination or estimation of unimolecular decomposition of *anti*-methacrolein oxide, see figure S3. It is not easy to conclude $k_{\text{uni}} < 50 \text{ s}^{-1}$ from this figure! At least authors should show a linear fit to the data and then use upper 2σ -uncertainty, not 1σ , to estimate k_{uni} . Closely related to figure S3, authors need to provide estimated CI concentrations; reader needs to have an idea of [*anti*-methacrolein oxide] for example in figure 2. Also, dependency of k_{uni} on precursor concentration would be needed. Please also show signal before $t=0$ in figure 2 to see any change in background due to photolysis. Another weakness is obtained $k_{\text{water-eff}} = (8 \pm 6) \times 10^{-17} \text{ cm}^3 \text{ s}^{-1}$ ($\pm 1\sigma$ uncertainty). Within $\pm 2\sigma$ uncertainty lower-limit value would be $-4 \times 10^{-17} \text{ cm}^3 \text{ s}^{-1}$! It is reviewer's opinion that authors have enough data for an upper limit (which is equally important!), but not for a bimolecular rate coefficient. Reviewer cannot understand why authors have not measured absolute absorption cross-section of their precursor, *E*-1,3-diiodo-2-methylprop-1-ene, in range important in this work. Authors set (!) and use $k_{\text{uni}} = 50 \text{ s}^{-1}$ in their calculation and discussion of atmospheric lifetime of *anti*-methacrolein oxide, although this value seems highly uncertain, as discussed above. More careful analysis is needed to obtain a reliable atmospheric lifetime of *anti*-methacrolein oxide.

In quite many places more careful proof-reading would have been in place.

Minor comments:

- line 36 are -> were
- line 42 "...fate of a CI would strongly depend on its structure." would sounds strange here
- Figure 3. It is practically impossible to see the Gauss fit.
- Line 195 – 196 ..(Figure 5).. -> (Figure 5 lower)
- Line 197. close to detection limit or close to measurement limit ?
- Lines 197 -199. "The highest [H₂O] used is ca. $6 \times 10^{17} \text{ cm}^{-3}$ (ca. 18 Torr), which is a larger portion of the bath gas if the total pressure is only 150 Torr (N₂ balance)." I do not understand this sentence.
- Line 222 Probably, actually especially, MACRO + MACRO reaction.
- Line 225 the plot of k_0 or k_{obs} ?
- Line 234 "This resonance-stabilization electronic structure is lacking in the alkyl CIs." Does not sound good. Probably "The alkyl-substituted CIs lack resonance-stabilized electron structure."
- Line 294. Explain a source of 0.5 in $0.5 \times 4 = 2$ expression.

Authors' point-by-point reply (in blue color) to Reviewers' comments

Reviewers' comments:

Reviewer #1 (Remarks to the Author):

The authors provide experimental data and supporting quantum chemical calculations establishing the rate constant for the reaction of anti methacrolein oxide (MACRO) with SO₂ and H₂O. Given the abundance of isoprene in the atmospheric and the slow unimolecular decay of anti MACRO, the reaction is highly significant. The authors have presented their results clearly and provide evidence of careful data analysis. This sound and fundamentally important study will advance atmospheric chemistry and merits publication. I have only two issues for the authors to consider:

(1) They should think more amply about the discrepancy between their MACRO + H₂O rate constant and the isoprene-derived CI + H₂O rate constant reported by Newland et al. (Phys Chem Chem Phys 2015). It is not enough to note that the majority of CI produced from isoprene ozonolysis is CH₂OO.

AUTHORS' REPLY:

We think the reviewer meant Newland et al., Atmos. Chem. Phys. 15, 9521–9536 (2015). The PCCP 2015 paper is not for isoprene-derived CIs. We have added a new section (copied below), Supplementary Note 2: Alternative simulation for Figure 4 of Newland et al., Atmos. Chem. Phys. 15, 9521–9536 (2015), to explain.

Supplementary Note 2: Alternative simulation for Figure 4 of Newland et al., Atmos. Chem. Phys. 15, 9521–9536 (2015).

We estimated the effect of the CI reactions with water vapor in the isoprene ozonolysis system in a manner similar to Newland et al. [Atmos. Chem. Phys. 15, 9521–9536 (2015)]. We separate the stabilized CIs produced in the isoprene ozonolysis into two groups, CH₂OO and C4-SCI (MVKO and MACRO). We found that we can fit the results of their Figure 4, but using a very different set of relative yields and rate coefficients (Model 1). The used kinetic parameters of our fit (Model 1) and those used by Newland et al. (Model 2, water-monomer-only scenario) are listed in Supplementary Table 3. See Supplementary Figure 5 for a comparison of the simulations.

Supplementary Table 3. Kinetic parameters used to fit the data of Figure 4 of Newland et al. Where $k_{\text{H}_2\text{O}}$, $k_{(\text{H}_2\text{O})_2}$, and k_{SO_2} are the rate coefficients of SCI reactions with water monomer, water dimer, and SO₂, respectively; k_{other} includes the unimolecular decay and other loss processes; ϕ is the relative yield of the SCI in the isoprene ozonolysis.

	SCI	$k_{\text{H}_2\text{O}} / \text{cm}^3 \text{ s}^{-1}$	$k_{(\text{H}_2\text{O})_2} / \text{cm}^3 \text{ s}^{-1}$	$k_{\text{SO}_2} / \text{cm}^3 \text{ s}^{-1}$	$k_{\text{other}} / \text{s}^{-1}$	ϕ
Model 1 (this work)	CH ₂ OO	$3.2 \times 10^{-16},^a$	$7.4 \times 10^{-12},^b$	$3.9 \times 10^{-11},^c$	15^d	0.77^d
	C4-SCI	$8 \times 10^{-17},^e$	$(1 \times 10^{-14})^f$	$3.9 \times 10^{-11},^g$	50^d	0.23^d
Model 2 ^h	CH ₂ OO	1.3×10^{-15}	0	3.9×10^{-11}	12	0.54
	C4-SCI	1.1×10^{-15}	0	3.9×10^{-11}	26	0.46

^a The best known rate coefficient in the literature by Berndt et al. Phys. Chem. Chem. Phys., 17, 19862–19873 (2015).

^b The best known rate coefficient in the literature by Smith et al. J. Phys. Chem. Lett., 6, 2708–2713 (2015).

^c The widely accepted rate coefficient in the literature by Welz et al. Science, 335, 204 (2012).

^d Fit to the data of Figure 4 of Newland et al. Atmos. Chem. Phys. 15, 9521–9536 (2015).

^e This work.

^f This value is too small to be sensitive to the fit.

^g Assumed the same as that for CH₂OO.

^h Copied from Table 2 of Newland et al. (water-monomer-only scenario).

Supplementary Figure 5. Upper: Comparison of the simulation results of Models 1 and 2. Both are fitted to the results of Figure 4 of Newland et al. Lower: Comparison of the f values (the fraction of the SCI produced that reacts with SO₂, as defined by Newland et al.) for CH₂OO and C4-SCI and their combined effect in the isoprene ozonolysis. The results of Model 1 are shown as solid lines; those of Model 2 as dashed lines.

This practice indicates that the complicated situation of isoprene ozonolysis can be interpreted with more than one set of kinetic parameters. Note that we do not have full knowledge about the experiments of Newland et al., thus, we should not claim our fitting can interpret their results. We just like to demonstrate that their situation is complicated and can have more than one interpretation.

We further plot the f value (the fraction of the SCI produced that reacts with SO₂, as defined by Newland et al.) in Supplementary Figure 5 (lower) to illustrate the effects of these two types of CIs. In Model 1, the pronounced decay of $f_{\text{CH}_2\text{OO}}$ with increasing [H₂O] reflects the fact that CH₂OO is quickly consumed by its reactions with water monomer and dimer. However, C4-SCI is consumed by the water reactions much slower. The effect of the remaining CH₂OO in the SCI + SO₂ reactions is insignificant for the range of [H₂O] > 1.5 × 10¹⁷ cm⁻³. As a result, the Y value (as defined in Newland et al.) would not increase in the same way when there is only CH₂OO. In fact, this situation has been proposed by Newland et al. as their possible explanation (iii), which is copied below.

(On page 9529-9530^{Newland et al., Atmos. Chem. Phys. 15, 9521–9536 (2015)}) “(iii) that multiple effects are affecting the curvature of the results shown in Fig. 4. Analogous plots for CH₃CHOO shown in Newland et al. (PCCP 2015) displayed a shallowing gradient with increasing [H₂O] (i.e. the opposite curvature to that caused by the (H₂O)₂ reaction). The probable

explanation for the curvature observed for CH_3CHOO is the presence of a mix of *syn* and *anti* conformers (Scheme 2) in the system and the competing effects of the different kinetics of these two distinct forms of CH_3CHOO . A similar effect may arise for the isoprene derived CRB-SCI which include multiple *syn* and *anti* conformers (see Scheme 2). The competition of this effect with that expected from the water dimer reaction may effectively lead to one masking the other under the experimental conditions applied.”

In addition, the temperature in the experiments of Newland et al. was between 287 and 302 K. This 15 K range may change the rate coefficients by a factor of 2, [Smith et al., J. Phys. Chem. Lett. 6, 2708–2713 (2015); DOI: 10.1021/acs.jpcllett.5b01109] resulting in extra complications.

Figure X1. Our simulation (magenta line) overlapped with those of Figure 4 of Newland et al.

We have also added the following sentences in the main text for clarification.

“At the time (2015) when the work of Newland et al. was published, the knowledge of the reaction kinetics of MVKO and MACRO was not available at all. As multiple CIs are produced in the isoprene ozonolysis system, the kinetics is rather complicated, especially when these CIs have very different reactivities towards water vapor. For example, CH_2OO , which has the predominant yield in the isoprene ozonolysis system,²⁻³ would be quickly consumed by its reaction with water vapor, but MVKO and MACRO would not. See Supplementary Note 2 for an alternative analysis to fit the data of Newland et al.⁶¹ In fact, Newland et al. have mentioned that the competing effects of the different kinetics of two distinct forms (*syn* and *anti* conformers) in the system may effectively lead to one masking the other under the experimental conditions applied.⁶¹”

(2) The authors should provide a little justification for the soundness of QCISD(T) theory for estimating reaction barriers and B3LYP for geometries and frequencies.

AUTHORS' REPLY:

We have added a new section “Justification for the theoretical methods” in Supplementary Note 3 to explain. The main points are summarized below.

- (i) We have confirmed that the QCISD(T) gives energies consistent with CCSD(T) and multireference methods for the $\text{CH}_2\text{OO}+\text{H}_2\text{O}$ reaction (see Lin et al. PCCP, 18, 4557, (2016)) and the CH_2OO unimolecular reaction (see Yin and Takahashi PCCP, 19, 12075 (2017)).
- (ii) We have confirmed that the B3LYP geometries of the TS for the $\text{CH}_2\text{OO}+\text{H}_2\text{O}$ and $\text{CH}_2\text{OO}+(\text{H}_2\text{O})_2$ reactions are similar to the QCISD(T) geometries (Table S5 of Lin et al. PCCP, 18, 4557, (2016)) and that this difference in geometry by these two quantum chemistry methods can induce an error of $0.3 \text{ kcal mol}^{-1}$ in the TS energies.
- (iii) The effect of low frequency vibrational modes has been checked to be minor, $\sim 0.1 \text{ kcal mol}^{-1}$ in the free energy difference. The effect of tunneling is not large, \sim a factor 1.2.

Reviewer #2 (Remarks to the Author):

The manuscript describes an interesting work from a well-known group on atmospheric-relevant kinetics of *anti*-methacrolein oxide (MACRO), which is resonance-stabilized that often leads to an increased radical stability and potentially lower reactivity. To say that, this reviewer feels that authors have over-interpreted or overused their not-so-accurate results too much.

AUTHORS' REPLY:

To the best of our knowledge, this work is the first kinetic investigation of methacrolein oxide (MACRO) with a direct detection method. We have performed additional measurements, redone our analysis and interpretations, and fixed some confusing (over-interpreted) sentences. The experimental and computational results, including their uncertainties, have been better assessed.

Although there are still some uncertainties in the experimental and computational results, our data are sufficient to (i) provide a much better estimate for the atmospheric lifetime of MACRO and (ii) tell which reaction is the predominant one among competing reactions. We believe these contributions are substantial. Please see below for our point-by-point reply.

This reviewer thinks that probably the main deficiency of this work is rather weak determination or estimation of unimolecular decomposition of *anti*-methacrolein oxide, see figure S3. It is not easy to conclude $k_{uni} < 50 \text{ s}^{-1}$ from this figure! At least authors should show a linear fit to the data and then use upper 2σ -uncertainty, not 1σ , to estimate k_{uni} .

AUTHORS' REPLY:

It is very difficult to experimentally determine the very slow unimolecular decay of *anti*-methacrolein oxide (*anti*-MACRO). More difficulty arises from the fact that the UV absorption band of MACRO overlaps with that of IO. Thus, we have to use the whole spectrum (instead of single-wavelength measurements) to separate the absorption contributions of *anti*-MACRO and IO. We have performed additional measurements and analysis (a lot of works), and replotted Supplementary Figure 3 (copied below). Following the reviewer's suggestion, we have shown a linear fit to the data and use the 2σ uncertainty. The intercept of this plot, which is $20 \pm 28 \text{ s}^{-1}$ ($\pm 2 \text{ se}$, se is the standard error of the intercept), gives a preliminary estimate of the unimolecular decomposition rate coefficient of MACRO.

Supplementary Figure 3. Upper: The observed decay rate coefficient k_{obs} plotted against the initial concentration of MACRO, $[\text{MACRO}]_0$ (derived by using the peak cross section of $3 \times 10^{-18} \text{ cm}^2$ reported by Vansco et al.³) when no other reactant was added. Experimental conditions: $P = 150\text{--}503 \text{ Torr}$; $P_{\text{O}_2} = 5\text{--}21 \text{ Torr}$; $T = 298 \text{ K}$; $[\text{precursor}] = (1.6\text{--}7.2) \times 10^{13} \text{ cm}^{-3}$ (using the cross sections measured in this work); $I_{248\text{nm}} = 0.90\text{--}3.88 \text{ mJ cm}^{-2}$. The intercept of this plot gives a preliminary estimate of the unimolecular decomposition rate coefficient of MACRO, which is $20 \pm 28 \text{ s}^{-1}$ ($\pm 2 \text{ se}$, se is the standard error of the intercept). Note that it is very difficult to measure this very small unimolecular decay rate coefficient for *anti*-MACRO, of which the absorption spectrum is very broad and overlaps with that of IO. Here we can only conclude that our experimental results are consistent with the theoretical predictions listed in Table 1.

Lower: k_{obs} plotted against the product of the initial concentration of the precursor and the photolysis laser fluence, $[\text{precursor}] \times I_{248\text{nm}}$. A clear linear dependence can be observed in this plot. The data are color-coded for various levels of $I_{248\text{nm}}$ and we can see that there is no clear correlation of k_{obs} with $I_{248\text{nm}}$ alone. That is, there would also be no clear correlation of k_{obs} with $[\text{precursor}]$ alone, since the horizontal axis is their product ($[\text{precursor}] \times I_{248\text{nm}}$).

We agree that the experimental uncertainty in k_{uni} is not very small. Therefore, we use the unimolecular rate coefficient from high-level ab initio theoretical calculations to estimate the atmospheric decay rate coefficient k_{atm} . The **justification** of using the theoretical value of k_{uni} is given in the revised main text (copied below).

“Best estimation of k_{uni} . It is very difficult to experimentally measure the very slow rate of the *anti*-MACRO unimolecular reaction. While our preliminary experimental data (Supplementary Figure 3) suggest that the unimolecular reaction is slow, we cannot nail down the value of k_{uni} by the experimental results. On the theoretical side, the unimolecular reaction of *anti*-MACRO proceeds through the OO bending channel forming dioxirane,^{32, 34, 39} similar to that of CH_2OO .^{41, 62} By comparing with the results of high-accuracy extrapolation protocols like HEAT-345(Q)⁶² or high-level multireference methods like

MRCI+Q (Davidson correction)/CBS,⁴¹ Yin and Takahashi have found that the QCISD(T)/CBS method slightly underestimates the barrier height of this channel (by ca. 0.4 or 1.2 kcal mol⁻¹, respectively) for CH₂OO.⁴¹ Our analysis in Supplementary Note 3 shows that for the MACRO unimolecular reaction, the electronic energy obtained by QCISD(T)/CBS would consistently underestimate the barrier height and other factors in the rate calculation like hindered-rotor partition function calculation and tunneling correction have very minor effects compared to that of the electronic energy. Therefore, our theoretical value (25 s⁻¹) of k_{uni} of *anti*-MACRO would only represent an upper limit.

Assuming such underestimation in the barrier heights (0.4 or 1.2 kcal mol⁻¹) is similar for the unimolecular reactions of MACRO and CH₂OO, we may have an overestimation of a factor of 2 or 7 for the reaction rate coefficient at 298 K. Thus, we think the best estimated k_{uni} at 298 K would be ca. $25/(2 \times 7)^{0.5} = 7 \text{ s}^{-1}$ (the uncertainty may be up to a factor of 3), which is (almost) coincident with the theoretical value of 10 s⁻¹ by Vereecken et al. (claimed uncertainty is ca. a factor of 5 for non-H-migration reactions).³² Although the uncertainty is still not very small, "for many assessments, however, it is sufficient to determine whether the reaction is significantly faster or slower than competing reactions", mentioned by Vereecken et al.³²”

It is important to note that our theoretical method would not overestimate the reaction barrier for the dioxirane channel, and thus, our theoretical value of k_{uni} (25 s⁻¹) is an upper limit.

Please also note that a smaller basis set (aug-cc-pVTZ) was used in the previous computational works (e.g., Anglada et al.) for the rate coefficients of MACRO reactions with water monomer and dimer, causing severe overestimation in the rate coefficients. Our results indicate that a complete basis set extrapolation (with QCISD(T) or CCSD(T)) is crucial in describing the reactions of MACRO with water monomer and dimer.

Closely related to figure S3, authors need to provide estimated CI concentrations; reader needs to have an idea of [*anti*-methacrolein oxide] for example in figure 2.

AUTHORS' REPLY:

We have used the reported peak absorption cross section of *anti*-MACRO of $3 \times 10^{-18} \text{ cm}^2$ by Vansco et al. [*J. Am. Chem. Soc.* **141**, 15058-15069 (2019)] to estimate the concentrations of *anti*-MACRO. The related Figures have been replotted.

Also, dependency of k_{uni} on precursor concentration would be needed.

AUTHORS' REPLY:

Please see above Supplementary Figure 3 (lower): “ k_{obs} plotted against the product of the initial concentration of the precursor and the photolysis laser fluence, [precursor] $\times I_{248\text{nm}}$. A clear linear dependence can be observed in this plot. The data are color-coded for various levels of $I_{248\text{nm}}$ and we can see that there is no clear correlation of k_{obs} with $I_{248\text{nm}}$ alone. That is, there would also be no clear correlation of k_{obs} with [precursor] alone, since the horizontal axis is their product ([precursor] $\times I_{248\text{nm}}$).” Thus, there is no clear dependency of k_{uni} on [precursor].

The Figure below shows the residue of the fit of Supplementary Figure 3 (upper) plotted against [precursor]. Again no clear dependency on [precursor] can be found.

Figure X2. Residue of the fit of Supplementary Figure 3 (upper) plotted against [precursor].

Please also show signal before $t=0$ in figure 2 to see any change in background due to photolysis.

AUTHORS' REPLY:

Figures 2, 4 and 5 have been replotted to show the signal before $t = 0$.

Another weakness is obtained $k_{\text{water-eff}} = (8 \pm 6) \times 10^{-17} \text{ cm}^3 \text{ s}^{-1}$ ($\pm 1\sigma$ uncertainty). Within $\pm 2\sigma$ uncertainty lower-limit value would be $-4 \times 10^{-17} \text{ cm}^3 \text{ s}^{-1}$! It is reviewer's opinion that authors have enough data for an upper limit (which is equally important!), but not for a bimolecular rate coefficient.

AUTHORS' REPLY:

Yes, we don't have confidence for the lower limit of the $k_{\text{water-eff}}$. Please note that a smaller value of $k_{\text{water-eff}}$ would suggest a longer atmospheric lifetime, which is consistent with our conclusion.

We have performed more measurements, improved our analysis (again, a lot of works), and revised the related text as below.

“Finally we chose the weighted average from six experimental sets (300 and 500 Torr and 298 K, Supplementary Table 2), to report the effective rate coefficient for the reaction of MACRO with water vapor, $k_{\text{water-eff}} = (9 \pm 5) \times 10^{-17} \text{ cm}^3 \text{ s}^{-1}$ ($\pm 1\sigma$). As mentioned above, we are not confident enough to determine the lower limit of $k_{\text{water-eff}}$.”

Our best estimate of k_{atm} at RH=70% and 298 K is 59 s^{-1} and a more conservative estimate is $k_{\text{atm}} < 74 \text{ s}^{-1}$ (if taking our theoretical upper limit of 25 s^{-1} for k_{uni}). These values are significantly smaller than previous estimates of 3900 or 350 s^{-1} (even after including the 2σ experimental error of 54 s^{-1} ($\sigma = 27 \text{ s}^{-1}$)). If using both upper bounds of k_{uni} and $k_{\text{water-eff}}$, we would have $k_{\text{atm}} < 128 \text{ s}^{-1}$, suggesting a bit lower steady-state [MACRO]. Nonetheless, the impact of MACRO on the oxidation of SO_2 would still be substantial due to the very large reaction rate coefficient of MACRO +

SO₂ reaction. We have revised the related text as below.

“Taking the best estimated k_{uni} (7 s^{-1}) and our experimental data of $k_{\text{water-eff}}$ (Table 1), we have $k_{\text{atm}} = 56 \text{ s}^{-1}$ (or $< 74 \text{ s}^{-1}$, if taking our theoretical upper limit of 25 s^{-1} for k_{uni}) for *anti*-MACRO at RH = 70% and 298 K. Note that the water reaction may still predominate in the decay processes of atmospheric *anti*-MACRO under humid conditions that are typical for tropical forests where the isoprene emission is large. And this atmospheric lifetime (ca. 18 ms, best estimate) is much longer than previously thought (0.3 or 3 ms, see Table 1), indicating that the atmospheric concentration of *anti*-MACRO would be much higher than previously expected. If using the upper limits of k_{uni} (25 s^{-1}) and $k_{\text{water-eff}}[\text{H}_2\text{O}]$ ($49+54=103 \text{ s}^{-1}$, 2σ upper bound, at RH = 70% and 298 K), we then have $k_{\text{atm}} < 128 \text{ s}^{-1}$, which would correspond to a lifetime longer than 7.8 ms.”

Reviewer cannot understand why authors have not measured absolute absorption cross-section of their precursor, *E*-1,3-diiodo-2-methylprop-1-ene, in range important in this work.

AUTHORS' REPLY:

The vapor pressure of the precursor, *E*-1,3-diiodo-2-methylprop-1-ene, is too low to quantify its number density. We quantified the consumed precursor by its weight loss and estimated its number density by using the integrated volume flow rate of the dilution gas as in our previous works [*Chem. Phys. Chem.* **21**, 2056-2059 (2020); *J. Chin. Chem. Soc.* **67**, 1563-1570 (2020)]. The result of the measured absolute absorption cross section is given in Supplementary Figure 4, which is copied below.

Supplementary Figure 4. The standard I_2 (upper panel) and IO (middle panel) spectra derived from our experimental spectra to include the instrumental functions (shown as blue lines). They are consistent with the reported spectra in the literature (black lines).^{7,8} The absolute absorption cross sections of the precursor, 1,3-diiodo-2-methylprop-1-ene, measured in this work are shown in the lower panel. The peak cross section was determined to be $(3.0 \pm 0.2) \times 10^{-17} \text{ cm}^2$ ($\pm 1 \text{ sd}$, from 8 measurements) at 238 nm.

Authors set (!) and use $k_{\text{uni}} = 50 \text{ s}^{-1}$ in their calculation and discussion of atmospheric lifetime of *anti*-methacrolein oxide, although this value seems highly uncertain, as discussed above. More careful analysis is needed to obtain a reliable atmospheric lifetime of *anti*-methacrolein oxide.

AUTHORS' REPLY:

As mentioned above, we do not use $k_{\text{uni}} = 50 \text{ s}^{-1}$ in our estimation and discussion of the atmospheric lifetime of *anti*-MACRO. When we mention " $k_{\text{uni}} \cong k_{\text{obs}}([\text{MACRO}]_0=0) < 50 \text{ s}^{-1}$, which gives a lifetime $> 20 \text{ ms}$ ", we like to show "This long-lived MACRO cannot be *syn*-MACRO. Thus, the observed signal should belong to *anti*-MACRO."

Instead, we use the theoretical results of $k_{\text{uni}} = 7 \text{ s}^{-1}$ (our best estimate) and $k_{\text{uni}} < 25 \text{ s}^{-1}$ (upper limit) in the revised manuscript with a solid justification (see above: “**Best estimation of k_{uni} ...**”) to estimate k_{atm} . Given the experimental uncertainty, we think it is more reliable to use the theoretical value in the case of the unimolecular reaction rate coefficient. This treatment is similar to the case for the rate coefficients of MVKO reactions with water vapor reported by Caravan et al. [Direct kinetic measurements and theoretical predictions of an isoprene-derived Criegee intermediate. *Proc. Nat. Acad. Sci. U. S. A.* **117**, 9733-9740 (2020)].

As given in the manuscript, we have performed a significantly better and careful analysis for k_{atm} . Please also note that even if taking $k_{\text{uni}} \leq 50 \text{ s}^{-1}$, the value of k_{atm} is still substantially smaller than previously thought.

In quite many places more careful proof-reading would have been in place.

Minor comments:

- line 36 are -> were

AUTHORS' REPLY: fixed.

- line 42 “...fate of a CI would strongly depend on its structure.” would sounds strange here

AUTHORS' REPLY: “the reactivity (thus the atmospheric fate) of a CI would strongly depend on its structure.”

- Figure 3. It is practically impossible to see the Gauss fit.

AUTHORS' REPLY: We have used a thicker red line to show the Gaussian fit.

- Line 195 – 196 ..(Figure 5).. -> (Figure 5 lower)

AUTHORS' REPLY: fixed.

- Line 197. close to detection limit or close to measurement limit ?

AUTHORS' REPLY: Should be “measurement limit”.

- Lines 197 -199. “The highest [H₂O] used is ca. $6 \times 10^{17} \text{ cm}^{-3}$ (ca. 18 Torr), which is a larger portion of the bath gas if the total pressure is only 150 Torr (N₂ balance).” I do not understand this sentence.

AUTHORS' REPLY: Sorry for not making it clear. The sentences have to revised to

“Note that the highest [H₂O] used is ca. $6 \times 10^{17} \text{ cm}^{-3}$ (ca. 18 Torr), which has replaced a larger portion (18/150 = 12%) of the bath gas if the total pressure is only 150 Torr (N₂ balance). Thus, we think it may require some cautions to view the data of 150 Torr, because the reaction environment (type of bath gas) changes at various [H₂O].”

- Line 222 Probably, actually especially, MACRO + MACRO reaction.

AUTHORS' REPLY: The related sentences have to revised to

“... while the intercept k_0 would account for other decay processes of MACRO that are independent on [SO₂], like reactions with radical byproducts (including MACRO self-reaction), unimolecular decay, etc.”

“This is due to the fact that the inevitable reactions of MACRO with radical byproducts, including I atoms, IO radicals, MACRO itself and the products from the fast decomposition of *syn*-MACRO (similar to the case of *anti*-MVKO)⁴² would shorten its lifetime.”

- Line 225 the plot of k_0 or k_{obs} ?

AUTHORS' REPLY: Sorry for the confusing. Should be k_{obs} .

- Line 234 "This resonance-stabilization electronic structure is lacking in the alkyl CIs." Does not sound good. Probably "The alkyl-substituted CIs lack resonance-stabilized electron structure."

AUTHORS' REPLY: The related sentences have to revised to

"... the extended conjugation of *anti*-MACRO correlates with the lower reactivity towards water vapor (monomer and dimer), since the alkyl-substituted CIs lack the resonance-stabilized electronic structure of the extended conjugation."

The above have all been fixed. We have done more careful proof-reading. Thanks a lot for your careful reading.

- Line 294. Explain a source of 0.5 in $0.5 \times 4 = 2$ expression.

AUTHORS' REPLY:

Sorry for the confusing expression. The relevant sentences have been revised to

"Overall, the oxidation rate of SO_2 by *anti*-MACRO would be larger than that by *syn*-MVKO by a factor of $([anti-MACRO]_{\text{ss}}/[syn-MVKO]_{\text{ss}})(k_{\text{SO}_2+anti-MACRO}/k_{\text{SO}_2+syn-MVKO})$. This factor would be larger than or close to unity, if we assume that these two CIs are mainly from isoprene ozonolysis with similar yields."

Reviewer #3 (Remarks to the Author):

Review of “Surprisingly long lifetime of methacrolein oxide, an isoprene derived Criegee intermediate, under humid conditions”

This study focuses on the determination of the reaction of the anti-MACRO SCI with water, SO₂ and its unimolecular loss rate. It is found that the effective reaction with H₂O is slower than previously reported from theoretical calculations which increases the lifetime of the anti-MACRO SCI from less than a ms to up to 20 ms. The reaction of the anti-MACRO SCI with SO₂ is also found to be very fast and can contribute to the formation of H₂SO₄ due to the longer lifetime of this SCI.

This study is well performed with a solid experimental and theoretical component. The results are of high interest for the community and I fully support the publication after the following points are considered.

Abstract. When talking about the water rate coefficient I would specify it is the effective one as highlighted further in the paper. I would also add the newly calculated lifetime and what was the previously thought value.

AUTHORS' REPLY: Revised (copied below) as suggested.

“However, the reaction of *anti*-MACRO with water vapor has been observed to be quite slow with an effective rate coefficient of $(9\pm 5)\times 10^{-17} \text{ cm}^3 \text{ s}^{-1}$ ($\pm 1\sigma$) at 298 K (300 to 500 Torr), which is smaller than current literature values by 1 or 2 orders of magnitude. Our results indicate that *anti*-MACRO has an atmospheric lifetime (best estimate ca. 18 ms at 298 K and RH = 70%) much longer than previously thought (ca. 0.3 or 3 ms), resulting in a much higher steady-state concentration.”

Introduction. At the end of the first paragraph, when discussing the importance of SCI as oxidants of SO₂ from previous studies it is important to distinguish between older studies which estimated a much larger SCI concentrations from more recent studies where a more complete body of reactions for SCI was included. At the end of the second paragraph also the study by Novelli et al. (2017) should be included as the first highlighting low expected steady state concentrations in the atmosphere.

AUTHORS' REPLY:

We have added the following sentences to highlight these issues.

“However, it is impractical to measure the very reactive CIs in the atmosphere. Their atmospheric concentrations can only be estimated through kinetics analysis. For example, Novelli et al. have given an average estimate of the CI concentration of ca. $5\times 10^4 \text{ molecules cm}^{-3}$ (with an order of magnitude uncertainty) for the two environments they studied.¹⁶ ... Note that older estimations may have larger uncertainties in the CI concentrations since the related reaction kinetics were not well determined at that time.”

On page 5 when discussing previous values for the reaction of anti-MACRO I would recommend to highlights the values from the study by Vereecken et al. (2017) as they are correctly accounting for the interconversion of the two conformers and apply the correct barrier height. Indeed, the difference between the measured k effective and the theoretical one from the study by Vereecken et al. (2017) is less than a factor of 10, well within the uncertainties.

AUTHORS' REPLY:

We have added the following sentences to highlight this issue.

“These theoretically predicted values are from Vereecken et al.³² who utilized the structure-activity relationships which considered the best available theoretical and experimental results.”

On page 6, in the last paragraph, similarly as for the abstract, I would specify what lifetime for MACRO was predicted in the atmosphere and from which studies.

AUTHORS' REPLY:

We have added the following sentences to highlight this issue.

“To our surprise, the reaction of MACRO with water vapor was found to be much slower than previous predictions by one or two orders of magnitude, implying much longer atmospheric lifetime (ca. 18 ms *vs.* 3 or 0.3 ms)^{32, 44} and higher steady-state concentrations for atmospheric MACRO.”

Table 1. I would add the experimental result for the unimolecular decomposition of anti-MACRO of $< 50 \text{ s}^{-1}$ as it is consistent with what found with the theory. I would also add the values from the study by Vereecken et al. (2017) for the *syn*-MVKO. In the caption below the table I would also add the water and water dimers concentrations. Is RH=75% and T=298K a representative set for isoprene dominated environments such as Amazonia? It could make sense to have two scenarios at higher and lower values and see the bracket of lifetime for the upper and lower limit.

AUTHORS' REPLY:

It is very hard to measure the very small unimolecular rate coefficient experimentally. See Supplementary Figure 3 for details. We can only say that the experimental results are consistent with the best theoretical predictions. Thus, we prefer not to add this experimental value on Table 1.

We have revised Table 1 to show a more complete set of data (including *syn*-MVKO) at two RH levels.

We have added the following sentences to highlight the high humidity environments.

“Note that the water reaction may still predominate in the decay processes of atmospheric *anti*-MACRO under humid conditions that are typical for tropical forests where the isoprene emission is large.”

Discussion. On page 16 it is underlined that the measured unimolecular must be smaller than 50 s^{-1} implying that the lifetime will be then larger than 20 ms. Although this is technically correct, it is not the full picture as the loss with water and water dimers contribute about 50% to the total loss of MACRO (even more if the unimolecular rate from the study by Vereecken et al. (2017), 10 s^{-1} , is used).

AUTHORS' REPLY:

We have added the following sentences to highlight this issue.

(page 20 now) “Note that the water reaction may still predominate in the decay processes of atmospheric *anti*-MACRO under humid conditions that are typical for tropical forests where the isoprene emission is large. And this atmospheric lifetime (ca. 18 ms, best estimate) is much longer than previously thought (0.3 or 3 ms, see Table 1), indicating that the atmospheric concentration of *anti*-MACRO would be much higher than previously expected.”

Atmospheric lifetime. Why is the value from the study by Vereecken et al. (2017) used and not the value as calculated within this study? Also, the theoretical value also carries some uncertainty, shouldn't also be considered in the calculation? And which studies reported a lifetime of less than 1 ms?

AUTHORS' REPLY:

We have performed a detailed analysis on the error sources of our theoretical estimation for the unimolecular rate. See Supplementary Note 3 for details. We also added a new section (copied below) in the main text to elaborate.

“**Best estimation of k_{uni} .** It is very difficult to experimentally measure the very slow rate of the *anti*-MACRO unimolecular reaction. While our preliminary experimental data (Supplementary Figure 3) suggest that the unimolecular reaction is slow, it cannot nail down the value of k_{uni} . On the theoretical side, the unimolecular reaction of *anti*-MACRO proceeds through the OO bending channel forming dioxirane,^{32, 34, 39} similar to that of CH₂OO.^{41, 62} By comparing with the results of high-accuracy extrapolation protocols like HEAT-345(Q)⁶² or high-level multireference methods like MRCI+Q (Davidson correction)/CBS,⁴¹ Yin and Takahashi have found that the QCISD(T)/CBS method slightly underestimates the barrier height of this channel (by ca. 0.4 or 1.2 kcal mol⁻¹, respectively) for CH₂OO.⁴¹ Our analysis in Supplementary Note 3 shows that for the MACRO unimolecular reaction, the electronic energy obtained by QCISD(T)/CBS would consistently underestimate the barrier height and other factors in the rate calculation like hindered-rotor partition function calculation and tunneling correction have very minor effects compared to that of the electronic energy. Therefore, our theoretical value (25 s⁻¹) of k_{uni} of *anti*-MACRO would only represent an upper limit.

Assuming such underestimation in the barrier heights (0.4 or 1.2 kcal mol⁻¹) is similar for the unimolecular reactions of MACRO and CH₂OO, we may have an overestimation of a factor of 2 or 7 for the reaction rate coefficient at 298 K. Thus, we think the best estimated k_{uni} at 298 K would be ca. $25/(2 \times 7)^{0.5} = 7 \text{ s}^{-1}$ (the uncertainty may be up to a factor of 3), which is (almost) coincident with the theoretical value of 10 s⁻¹ by Vereecken et al. (claimed uncertainty is ca. a factor of 5 for non-H-migration reactions).³² Although the uncertainty is still not very small, "for many assessments, however, it is sufficient to determine whether the reaction is significantly faster or slower than competing reactions", mentioned by Vereecken et al.³²”

If we take the $k_{\text{water-eff}}$ value by Anglada et al., the lifetime of MACRO would be ~0.3 ms. See Table 1.

Impact of anti-MACRO on the oxidation of atmospheric SO₂. Which other species, apart from isoprene, produce anti-MACRO?

AUTHORS' REPLY:

We are not experts on this issue. Isoprene is well known for its high abundance, such that the isoprene-derived CIs are important.

Conclusion. The last line is interesting, but I wonder what is the point of it? Either atmospherically relevant VOCs produce SCI which are long lived or, from an atmospherically point of view, the MACRO might well be the only “long” lived SCI. Which is very interesting on its own.

AUTHORS' REPLY:

We thank the reviewer for this comment. We have added a couple of sentences to make it clearer.

“As shown above and in the literature,³⁶ a resonance-stabilized electronic structure plays an interesting role for CIs. It reduces the reactivity of unimolecular decay and reactions with water vapor, but not for the reactions with SO₂. Thus, having a resonance-stabilized electronic structure may be a new direction for searching for a long lived CI that is able to oxidize atmospheric SO₂.”

References

Novelli, A., Hens, K., Tatum Ernest, C., Martinez, M., Nölscher, A. C., Sinha, V., Paasonen, P., Petäjä, T., Sipilä, M., Elste, T., Plass-Dülmer, C., Phillips, G. J., Kubistin, D., Williams, J., Vereecken, L., Lelieveld, J., and Harder, H.: Estimating the atmospheric concentration of Criegee intermediates and their possible interference in a FAGE-LIF instrument, *Atmos. Chem. Phys.*, 17, 7807-7826, doi:10.5194/acp-17-7807-2017, 2017.

Vereecken, L., Novelli, A., and Taraborrelli, D.: Unimolecular decay strongly limits the atmospheric impact of Criegee intermediates, *Phys Chem Chem Phys*, 19, 31599-31612, doi:10.1039/C7CP05541B, 2017.

REVIEWERS' COMMENTS:

Reviewer #1 (Remarks to the Author):

The authors have fully addressed both my concerns and those of the other two reviewers regarding the original version of the manuscript. I recommend that the current version be published without any further revision.

Reviewer #2 (Remarks to the Author):

I am satisfied with the authors' response (changes and additions to the original Manuscript and Supplementary Information) and suggest that the work can now be published.

Reviewer #3 (Remarks to the Author):

As the second version of the paper I think the Authors have reacted to my comments and I am satisfied with the current version of the manuscript.

I recommend publication.

REVIEWERS' COMMENTS:

Reviewer #1 (Remarks to the Author):

The authors have fully addressed both my concerns and those of the other two reviewers regarding the original version of the manuscript. I recommend that the current version be published without any further revision.

Reviewer #2 (Remarks to the Author):

I am satisfied with the authors' response (changes and additions to the original Manuscript and Supplementary Information) and suggest that the work can now be published.

Reviewer #3 (Remarks to the Author):

As the second version of the paper I think the Authors have reacted to my comments and I am satisfied with the current version of the manuscript.

I recommend publication.

AUTHORS REPLY:

Thank the reviewers.